# Phase Diagram of Pickering Emulsions Stabilized by Cellulose Nanocrystals

**DOI:** 10.3390/polym15132783

**Published:** 2023-06-22

**Authors:** Louise Perrin, Sylvie Desobry-Banon, Guillaume Gillet, Stephane Desobry

**Affiliations:** 1Laboratoire d’Ingénierie des Biomolécules (LIBio), Université de Lorraine, 2 Avenue de la Forêt de Haye, BP 20163, 54505 Vandœuvre-lès-Nancy Cedex, France; sylvie.desobry@univ-lorraine.fr (S.D.-B.); stephane.desobry@univ-lorraine.fr (S.D.); 2SAS GENIALIS Route d’Achères, 18250 Henrichemont, France; g.gillet@genialis.fr

**Keywords:** caprylic/capric triglycerides, stability, phase diagram, mechanisms, biomolecules

## Abstract

Cellulose is a promising renewable and biocompatible biopolymer for stabilizing Pickering emulsions (PEs). In the present study, PEs were produced by low-frequency ultrasounds with cellulose nanocrystals (CNCs) and caprylic/capric triglycerides. Phase diagrams allowed to understand mechanisms of formation and long-term stabilization of PEs. Emulsion type, continuous phase viscosity, and yield of oil incorporation were studied after PEs formation. Droplet size, oil release, and stability were measured weekly up to 56 days of storage. Results showed that oil mass fraction above 70% *w*/*w* led to unstable W/O PEs. Lower oil mass fraction formed O/W PEs of stability depending on CNC content and oil mass fraction. Droplet size stability increased with CNCs/oil ratio. A very low CNCs/oil ratio led to phase separation and oil release. High CNC content stabilized oil droplets surface, increased aqueous phase viscosity, and prevented creaming. Highly stable PEs were produced for CNC content above 3% (*w*/*w*) and oil mass fraction below 50% (*w*/*w*). Mechanisms for PEs formation and stabilization were proposed for various CNC contents and oil mass fractions.

## 1. Introduction

Emulsions are used daily in many fields, such as food, skin care and drugs, but also in agriculture, paint and bitumen industries, or chemistry [1,2].

Emulsions stabilization is reached by surface active compounds, which may unfortunately have adverse effects on human health and environment, especially those of synthetic origin [3,4,5,6]. As health and environment concerns are becoming priorities for consumers, greener products are formulated with natural ingredients. Industries invest in new ways to stabilize emulsions such as less toxic, biocompatible, and biodegradable bioemulsifiers [5,7]. Another way consists of using colloidal solid particles to produce emulsions called Pickering Emulsions (PEs) [8].

In PEs, solid particles are located at oil/water interface to form a mechanical barrier and avoid coalescence. Irreversible adsorption of solid particles is progressive during emulsification. Particles form a three-dimensional network between droplets, which slows down their displacement, and improve emulsion stability [9].

Among all PEs preparation methods, such as high-pressure homogenization, microfluidization, or microfluidic, ultrasound emulsification is mostly used [10,11]. Indeed, the cavitation phenomenon, specific to the ultrasound source, acts efficiently for PEs formation. Cavitation forces favor solid particles dispersion by overcoming energy barriers, and enable solids to adsorb at oil/water interface [12].

PEs are generally stable and their obtention depends on a few main parameters. One of them is the solid particle wettability measured by contact angle (θ) at particle/water/oil interface. When θ is less than 90°, an oil-in-water type emulsion is produced, whereas if θ is greater than 90°, a water-in-oil emulsion is obtained. Moreover, a contact angle below 15° or above 165° leads to an unstable emulsion [13]. Solid particle size, shape, and charge influence PEs formation and stability. Solid particles must be smaller by at least one order of magnitude than droplet size [14], except for fibrillated particles, which can bend at the oil/water interface [10]. The larger the particles, the more energy is required for their adsorption on oil/water interface [11]. Moreover, various shapes can be used as sphere, rod, fiber, cube, ellipse, nanotube, as well as disk [15,16]. The electric charge plays a role in particle aggregation or repulsion at oil/water interface, and influences PEs formation. Particle charge also impacts droplet electrostatic repulsion, and, therefore, emulsions stability [17].

Many biomaterials can be used to stabilize PEs [18], among them cellulose is particularly promising [19]. Cellulose, the most abundant biopolymer on Earth, is renewable, biocompatible, and widely available. It can be extracted from wood, plants, marine animals, and even from some bacteria [20]. In addition, this biopolymer is safe for health and environment, due to its biodegradability and non-toxicity. All of these advantages explain why cellulose is increasingly studied and used to replace petroleum-based ingredients to produce greener products [21].

PEs can be obtained with cellulose in form of fibrils or micro- or nano-sized crystals. Cellulose nanofibrils (CNFs) and cellulose nanocrystals (CNCs) represent the highest use [19,22,23]. PEs formation and stability can be influenced by CNC content. For example, Kasiri et al. (2018) formulated PEs with CNCs extracted from pistachio shells and corn oil. They showed that droplet size decreased with increasing CNC content, from around 17 µm for 0.1% CNCs down to 2 µm for 1.5% CNCs. PEs stability was better with higher CNC content [24]. These results have been confirmed by numerous others studies [25,26,27,28,29]. On the contrary, increasing oil mass fraction favored larger oil droplets formation and poorer PEs stability [28,30,31]. Ionic strength, pH, oil type, and CNCs nature can also have an impact on PEs properties [25,29,32,33,34].

In this study, PEs composed of water, CNCs, and Miglyol^®^ 812N (caprylic/capric triglycerides) were emulsified by low-frequency ultrasounds. The objective was to study effects of CNC content and oil mass fraction on PEs formation and stability. As few phase diagrams on PEs are available in the literature, this representation was chosen to illustrate PEs structure and stability over a wide range of CNCs and oil contents. Hypotheses about mechanisms of PEs formation and stability were formulated.

## 2. Materials and Methods

### 2.1. Materials

Cellulose nanocrystals (CNCs) were purchased from Celluforce (Montreal, QC, Canada). CNCs, produced from wood by acid hydrolysis using sulfuric acid, contain sulfated half-esters fixed on the hydroxyl groups [35]. Oil composed of caprylic/capric triglycerides (Miglyol^®^ 812N) was produced by IOI Oleochemical GmbH (Witten, Germany). This oil was chosen for its high purity, and for its common uses in cosmetics, as well as for its safety when used in pharmaceutical products [36]. Absence of surface active impurities in this oil type was demonstrated [37]. Deionized water used was produced with the Aquadem purification system (Veolia, Aubervilliers, France) to reach a resistivity of 18 MΩ·cm.

### 2.2. Phase Diagrams Definition and Emulsification Process

To study PEs formation and stability, formulations were made in deionized water with CNC amount varying from 0.5 to 5% (*w*/*w*) of total emulsion, and oil amount ranging from 5 to 90% (*w*/*w*) of total emulsion. Phase diagrams present only CNC content and oil mass fraction knowing that water complete the formulation to 100%. Total emulsion weight was fixed to 125 g. Thirty-six emulsions were produced from which only 27 will be presented for clarity on phase diagrams.

CNCs were suspended in water for 12 h at 20 °C under stirring before use. Hydrophilic phase (CNCs suspension) and hydrophobic phase (caprylic/capric triglycerides) were pre-emulsified by high-speed homogenization for 5 min using an Ultra-turrax^®^ T-25 equipped with a S 25N-18G dispersal tool (IKA-Werke, Staufen im Breisgau, Germany). Emulsification was then performed by low-frequency ultrasounds (20 kHz) using a Fisherbrand™ sonicator 500 W (Fisher Scientific, Waltham, MA, USA), with an amplitude of 40% and cycles of 25 s ON and 5 s OFF. Emulsion temperature was maintained at 20 ± 3 °C during ultrasound treatment in a thermostated reactor. The pH value of emulsions at the end of treatment was 6.0 ± 0.4.

### 2.3. Emulsion Type Determination

Emulsion type (O/W or W/O) was determined by drop dilution test [38]. A Pickering emulsion drop was added in water and in oil. The O/W emulsion was defined when the emulsion was diluted in water and formed two phases in oil. On the contrary, the W/O emulsion was defined when the emulsion formed two phases in water and was diluted in oil.

### 2.4. Yield of Oil Incorporation and Oil Release during Storage

To evaluate process efficiency on oil emulsification, oil and CNC content in emulsions was weighted after water evaporation at 45 °C for 12 h of 3.000 g of just prepared PE. Emulsified oil content was calculated by considering that the total initial CNC content was emulsified.

Yield of oil incorporation (% *w*/*w*) was defined by comparing initial oil amount introduced in emulsions and oil amount just after the emulsification process was achieved.

Oil release (% *w*/*w*) during storage was determined by comparing oil amount in emulsions after storage at 20 °C and oil amount present in fresh emulsions. Before determining oil content in emulsion, the samples were gently stirred manually to resuspend cream, as creaming is a reversible process that is not considered as an oil release. All assays were performed in triplicate.

### 2.5. Droplet Size Measurement

Droplet size distribution of PEs was measured by laser light scattering using a particle size analyzer, Malvern Mastersizer 3000 (Malvern instruments Ltd., Malvern, UK) equipped with a Hydro MV wet dispersion unit and blue and red light sources at 470 and 632.8 nm, respectively. This system enables to measure droplet size between 0.01 and 3500 µm. Samples were dispersed in water to reach an obscuration between 6% and 8%. Five measurements were made per sample. Size distribution was determined by fitting scattered light intensity measured by the different detectors to the Mie light scattering theory.

### 2.6. PEs Stability

PEs stability was determined by both visual observations and multiple light scattering measurements. These measurements were performed with a Turbiscan Classic MA2000 apparatus (Formulaction, Toulouse, France) using a pulsed near infrared light source at 850 nm. Seven milliliters of PEs samples were placed in a specific glass cell, which was then scanned by a light source. Detection was controlled by backscattering detector measuring light scattered by the sample at 45° to the incident beam. Measurements, performed in duplicate, were carried out after the emulsification process was achieved weekly, up to 56 days of storage.

### 2.7. Contact Angle Measurements

Static sessile-drop contact angle measurements were performed to determine CNC hydrophilicity. CNCs were first suspended in water for 12 h under stirring at 20 °C, then CNCs films were prepared by a casting method. Films were dried in an atmosphere at 20 °C and 40% relative humidity for 4 days, then conditioned in a desiccator containing silica gel and stored at 20 °C for at least 2 days before measurements. Contact angle with water was determined by dropping 3 µL of water on film surface. Measurement was performed 2 s after deposit by using Dino-lite High Magnification digital microscope and DinoCapture 2.0 software (AnMo Electronics Corporation, Taiwan, China).

### 2.8. Rheological Measurements

Rheological measurements were performed with a discovery hybrid rheometer, HR20 (TA Instruments, New Castle, DE, USA) using a 50 mm parallel plate geometry and a gap height between the two plates of 250 µm. Before starting measurement, an equilibration time was defined as 180 s. Viscosity measurements were performed in steady-state mode and in the shear rate range of 0.01 to 1000 s^−1^, at 20 ± 0.1 °C. Measurements were carried out 1 day after CNC suspensions preparation. Data obtained were fitted using Carreau–Yasuda model.

### 2.9. Statistical Analysis

Data were analyzed by one-way analysis of variance (ANOVA), using RStudio software, version 1.3.959 (R-tools Technology, Richmond Hill, ON, Canada). Differences at *p* < 0.05 were considered significant.

## 3. Results

Formation and stability of PEs composed of CNCs and caprylic/capric triglycerides were studied by constructing phase diagrams.

### 3.1. PEs Formation

From visual observations, dilution method and yield of oil incorporation, four different Pickering emulsions categories were defined (Figure 1a). Category I corresponds to homogeneous white O/W Pickering emulsions (PEs-I). Category II corresponds to biphasic system with a white O/W Pickering emulsion and more than 5% (*w*/*w*) of the oil initially added that was not emulsified (PEs-II). Category III includes coarse homogeneous W/O Pickering emulsions (PEs-III), and category IV refers to biphasic system with coarse W/O Pickering emulsions, and a CNCs precipitate to the bottom of the emulsions (PEs-IV).

All PEs are presented in a phase diagram (Figure 1b). When oil mass fraction was less than 40% (*w*/*w*), and regardless content of CNCs, O/W PEs-I were obtained. Higher contents of oil were stabilized for high CNC content (PEs-I) or gave biphasic emulsion (PEs-II), showing that a minimum CNCs/oil ratio was necessary to formulate homogeneous O/W PEs, as found by Kasiri and Fathi (2018) [24]. Contrary to some studies, very high oil content (above 70% *w*/*w*) did not enable to obtain high internal phase Pickering emulsions [39,40], but gave inverse W/O emulsions, i.e., PEs-III for low CNC content (less than 2% *w*/*w*) and PEs-IV when high content of CNCs precipitated (from 2 to 5% *w*/*w*). These W/O PEs were coarse due to the low contact angle found for CNC with water (32 ± 3°) characterized CNC as hydrophilic in nature and able to stabilize O/W PEs, but not W/O PEs, according to Finkle’s rule [41].

Kalashnikova et al. (2012) showed that it was not possible to produce PEs with sulfated CNCs, except by adding salt to screen electrostatic forces and, thus, suppress repulsive interactions between CNCs [42]. However, our results showed that O/W PEs formulated with sulfated CNCs, and without salt addition, were obtained over a large range of oil content. Recent studies showed that many physical properties related to oil (viscosity, polarity, solubility, and density) and CNCs (size, shape, surface charge, and wettability) can positively influence PEs [25,43,44], despite of sulfated CNC charge.

### 3.2. PEs Stability

PEs stability was studied for up to 56 days of storage by observing visual aspect, measuring oil release, droplet size, and creaming phenomenon. Two examples of O/W PEs are given in Figure 2 and Figure 3. The first one corresponds to a homogeneous emulsion stored for up to 56 days that contained 4% (*w*/*w*) CNCs and 50% (*w*/*w*) oil (Figure 2a). As expected, multiple light scattering measurements indicated no variation of backscattering data as a function of time (Figure 2b), showing that no destabilization phenomena, such as creaming or coalescence, occurred in this PE.

PE with creaming during storage is presented in Figure 3. Multiple light scattering measurements confirmed this destabilization with a decrease of backscattering signal at the bottom of emulsion, indicating a clarification phenomenon and an increase of backscattering signal at the top of emulsion highlighting the creaming. However, in contrast to results previously reported by Dias Meirelles et al. (2020), no increase of backscattering signal was observed at the bottom of the emulsion, showing that CNCs did not precipitate [30] (Figure 3b). During storage, serum layer thickness increased, revealing slow oil droplets up movement. This creaming phenomenon was reversible, and manual homogenization resulted in a white and homogeneous PE.

These emulsions are representative of two types of PEs over the four categories distinguished during 56 days of storage (Figure 4), i.e., PEs-A for stable homogeneous PEs and PEs-B for biphasic system with serum and cream layers. PEs-C corresponds to triphasic systems with an oil phase containing more than 5% (*w*/*w*) of initially emulsified oil, then cream and serum layers. A phase separation defined PEs-D with a large oil layer and an aqueous suspension of CNCs that tends to precipitate.

Using this classification, phase diagrams were made after 7, 14, 28, and 56 days of storage (Figure 5). The homogeneous O/W PEs-I (Figure 1) gave PEs-A and PEs-B during storage. For a CNC content greater than 3% (*w*/*w*), PEs-A were stable without creaming while a CNC content lower than 3% (*w*/*w*), creaming occurred during storage for PEs-B. Bai et al. (2019) did not observed creaming for PEs containing 2% (*w*/*w*) CNCs and 10% (*w*/*w*) corn oil after 14 days of storage [32]. These stable PEs obtained for a lower CNC content than in our study could be explained by salt addition (0.06% *w*/*w*). Assuming that CNCs are adsorbed at oil/water interface [32,45], the oil release observed during storage for O/W PEs-C indicated that CNC content was insufficient to stabilize all the oil content. W/O PEs-III and PEs-IV were not stable with rapid phase separation (PEs-D), occurring within hours of their formulation, due to the hydrophilic character of CNCs.

From seven to 56-days storage, few changes were observed for all O/W PEs (types A, B, and C) except in the junction zone between them. In this critical region, the instability of PEs increased from PEs-A to PEs-B and from PEs-B to PEs-C. PEs-D corresponded to all unstable W/O PEs (from domains III and IV in Figure 1b). Contrary to Meirelles et al. (2020), who observed oil release for PEs containing only 7.5% (*w*/*w*) oil and 0.5 or 1% (*w*/*w*) CNCs [30], in our study, no oil release was observed after 56 days of storage for PEs containing up to 30% (*w*/*w*) oil. The better stability observed in our study may be due to different CNC origin or oil composition. Thus, CNCs from wood and caprylic/capric triglycerides appear to be promising for stabilizing EPs over a long-term storage.

### 3.3. Study of Mechanisms Acting on O/W PEs-I Formation and Stability

Properties of O/W PEs-I were investigated to understand the creaming mechanisms observed during storage. Creaming phenomenon is a gravitational mechanism described, in a simplified way, by Stokes’ law (Equation (1)):(1)$$v_{\mathrm{creaming}}=-\frac{2\;g\;r^{2}\;\Delta \rho }{9\;\eta_{1}},$$
where $v_{\mathrm{creaming}}$ is the creaming velocity (m·s^−1^), $\Delta_{\rho }$ the density difference between continuous and dispersed phases (kg·m^−3^), g the gravitational constant (m·s^−2^), r the droplet radius (m), and $\eta_{1}$ the continuous phase viscosity (Pa·s) [14]. According to Equation (1), creaming velocity increases with droplet size, but decreases as continuous phase viscosity increases.

Granulometric profiles for O/W PEs- I emulsions gave bimodal distribution (Figure 6) as observed by Chen et al. (2019) [46]. High CNCs/oil ratio corresponded to submicronic droplets (Figure 6b,c), while low CNCs/oil ratio resulted in micrometric oil droplets (Figure 6a,d). This reduction in droplet size with increasing CNCs/oil ratio was consistent with the literature [10,47].

Droplet coalescence occurred in PE with the lowest CNCs/oil ratio (equal to 0.05 with 0.5% (*w*/*w*) CNCs and 10% (*w*/*w*) oil) (Figure 6a). This suggests that the CNC content was not sufficient to stabilize oil/water interface created during emulsification. However, subsequent coalescence was limited and gave micrometric size droplets without significant oil release during storage (less than 5% (*w*/*w*) of emulsified oil). Droplets reorganized after their formation to reach an equilibrium state between CNCs available and specific oil droplet surface. As CNC content was a limiting factor to stabilize the smaller initial droplets, they coalesced, reducing the specific interfacial area, and increasing possibility for CNCs to cover droplets surface. Increase in droplet size occurred until sufficient surface coverage was achieved to stop coalescence. This phenomenon is called “limited coalescence” [48].

For PE containing 10% (*w*/*w*) oil and higher CNC content (4% *w*/*w*), droplet size distributions remained stable during storage (Figure 6b). This could be explained by higher CNCs/oil ratio equalled to 0.4. A part of CNCs would adsorb at oil/water interface and CNCs in excess would remain in aqueous phase to form a three-dimensional network between droplets, preventing them from coming together and then coalescing [9]. This PE remained homogeneous without creaming during storage showing little movement of droplets from multiple scattering measurements (data not shown). The higher CNC content could also form CNC multilayers at the oil/water interface [49], and avoided coalescence and Ostwald ripening. Further analysis on local organization/aggregation of CNCs at interface would be useful to study these phenomena. Among all studied PEs, stable droplet without coalescence during storage were obtained when the CNCs/oil ratio was greater than 0.09.

The same CNCs/oil ratio equal to 0.4 (2% (*w*/*w*) CNCs and 5% (*w*/*w*) oil) gave similar droplet size distribution (Figure 6c compared to Figure 6b). However, stability was different with slight creaming for PE containing less dispersed phase. A difference in stability for the same CNCs/oil ratio has already been observed in literature [30].

With high oil mass fraction (45% *w*/*w*) and CNC content (4% *w*/*w*), droplet size was micrometric and a slight increase was noticed during storage up to 56 days, but PE remained stable (Figure 6d). The absence of creaming was explained by the high CNCs-stabilized dispersed phase content and the high viscosity of continuous phase that minimized creaming velocity in Stokes’ law (Equation (1)). The viscosity of aqueous phase containing 1% (*w*/*w*) to 5% (*w*/*w*) CNCs was measured (Figure 7). Increase in shear rate caused a decrease in viscosity of each CNCs suspension, showing that CNCs suspensions presented a shear-thinning behavior. Viscosity also depended on CNC concentration. The higher the CNC content, the higher the viscosity. Thus, as CNC content increased in PEs, continuous phase viscosity increased, which promoted PEs stability by decreasing creaming velocity.

For PEs containing 2% (*w*/*w*) CNCs, and a moderate content of oil (25% (*w*/*w*)), reversible creaming was observed (Figure 8). After 56 days of storage, two main droplet populations were present with the submicronic particles in the serum layer and the larger micrometric droplets in cream, confirming that larger droplets creamed faster than smaller ones. Phase separation was reversible after homogenization, indicating that CNCs still stabilized the O/W interface. Submicrometric particles were not detected in homogenized emulsion, probably due to their small proportion compared to micrometric droplets.

### 3.4. Hypothesis of Mechanisms Involved in PEs Formation and Stability

According to our results, different mechanisms could be proposed for PEs formation and stability, depending on CNC content and oil mass fraction. When oil mass fraction is low, PEs stability is mainly due to CNC content (Figure 9a). Low CNC content is insufficient to stabilize droplets interface, so limited coalescence occurs, i.e., droplets coalesce until forming a continuous CNC layer at oil/water interface. If the CNC content is very low relative to oil mass fraction (very low CNCs/oil ratio), PEs may even be unstable with large oil release (not represented on figure). When CNC content increases, initial surface coverage of droplets become sufficient to prevent coalescence and droplets are stabilized by Pickering mechanism, but creaming still occurs until continuous phase viscosity increases for high CNC content. This increase in viscosity would be due to CNCs in excess remaining in continuous phase. Thus, when oil mass fraction is low, PEs stability is primarily due to the continuous phase viscosity, governed by CNC content.

When oil mass fraction is high, PEs stability no longer depends solely on CNC content, but oil content also has an impact (Figure 9b). When CNC content is low, droplets are unstable with coalescence leading to oil release and phase separation. Even if droplets coalesce, surface coverage is still insufficient to stabilize them (CNCs/oil ratio too low). When CNC content increases, and thus also the CNCs/oil ratio, only slight coalescence occurs. Moreover, PEs remain homogeneous and stable, whereas for similar CNC content, but lower oil mass fraction, creaming occurred. Apparent viscosity of PEs has been shown to increase with oil mass fraction [46,50], which could be explained by interactions between CNCs from adjacent droplets [51]. These interactions are, therefore, probably favored by proximity between droplets due to high oil mass fraction. These interactions between droplets could limit their movement and prevent creaming. It will be interesting to study rheological properties of PEs to better understand phenomena and interactions involved in their stability. Finally, when CNC content is high, droplet surface is saturated by CNCs, which prevents coalescence, and CNCs in excess remain in disperse phase. That increases disperse phase viscosity, which reinforces PEs stability, in addition to interactions between droplets.

## 4. Conclusions

Formation and stability of PEs composed of water, CNCs (0.5 to 5% *w*/*w*), and caprylic/capric triglycerides (5 to 90% *w*/*w*) were studied. Phase diagrams were produced to gain an overview of PEs formation and stability over a wide range of CNC content and oil mass fraction. They allowed us to understand the effects of composition on PEs stability. These diagrams help formulation of PEs with CNCs and support the choice of PE composition according to desired stability.

Homogeneous O/W PEs were obtained up to oil mass fraction near 40% (*w*/*w*). For oil mass fraction between 40 and 70% *w*/*w*, PEs formation depended on CNC content. Above 70% (*w*/*w*) oil, PEs were W/O type, and were unstable during storage.

Among homogeneous O/W PEs initially formed, stability of PEs and droplet size depended on CNC content and oil mass fraction. During storage, increase in droplet size showed that CNC content was insufficient to form a CNCs layer at oil/water interface (CNCs/oil ratio too low). Increase of CNCs/oil ratio avoided coalescence but did not prevent creaming. Reduced creaming was reached through an increase in continuous phase viscosity due to CNCs excess, or interactions between CNCs particles of adjacent droplets. Emulsions were stabilized by CNCs adsorption at oil/water interface forming a solid barrier, increasing viscosity of continuous phase, and/or interactions between droplets with high oil mass fraction. These mechanisms are dependent on CNCs and oil contents. Further research is needed to confirm these hypotheses. CNCs organization in PEs could be studied by microscopy methods, such as confocal laser scanning microscopy. PEs rheological measurements could also provide information about interactions between CNCs. Effect of ultrasound treatment on CNCs should also be studied to ensure that this treatment does not affect CNCs’ integrity, or, on the contrary, would improve its properties to stabilize PEs. Moreover, as several studies showed that ionic strength and pH had an impact on PEs formation and stability, it could be interesting to study these parameters’ effects on PEs formulated with a wide range of CNCs and oil contents.

This study showed that PEs composed of unmodified CNCs and a model oil containing triglycerides, with high oil mass fraction (up to 50% *w*/*w*), can be stable (i.e., remain homogeneous, or with reversible creaming) during long-term storage. They could, therefore, be used in many fields such as in the food, cosmetic, or pharmacy industries to formulate greener products.

## Figures and Tables

**Figure 1 polymers-15-02783-f001:**
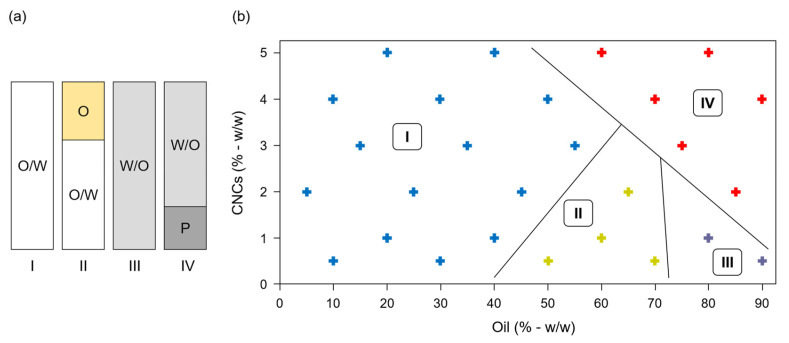
Pickering emulsions (PEs) formulated with cellulose nanocrystals (CNCs) and caprylic/capric triglycerides. (**a**) Classification of PEs obtained just after emulsification. O/W: oil-in-water emulsion; O: oil; W/O: water-in-oil emulsion; P: CNCs precipitate. (**b**) Phase diagram of the 27 freshly prepared PEs classified according to types I to IV. Blue, green, grey and red crosses represent PEs belonging to categories I, II, III and IV, respectively.

**Figure 2 polymers-15-02783-f002:**
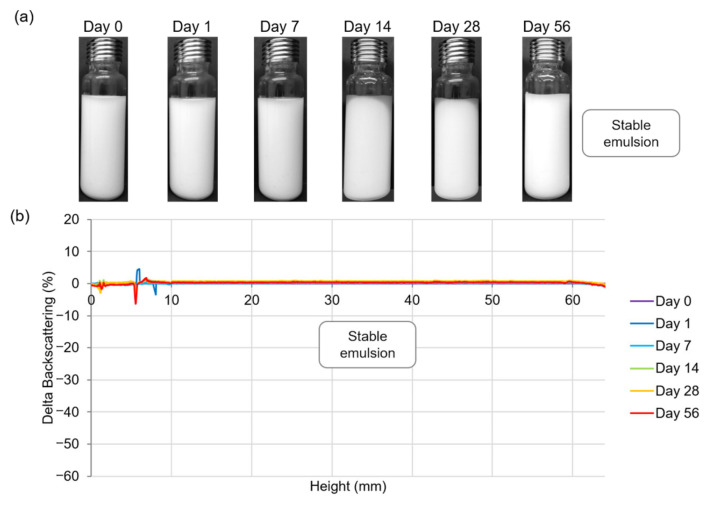
Stability evolution of Pickering emulsion composed of 4% (*w*/*w*) CNCs and 50% (*w*/*w*) oil. (**a**) Visual appearance of emulsion from 0 to 56 days of storage at 20 °C. (**b**) Multiple light scattering analysis showing backscattering data as a function of sample height (6 to 64 mm) and of storage time (0 to 56 days).

**Figure 3 polymers-15-02783-f003:**
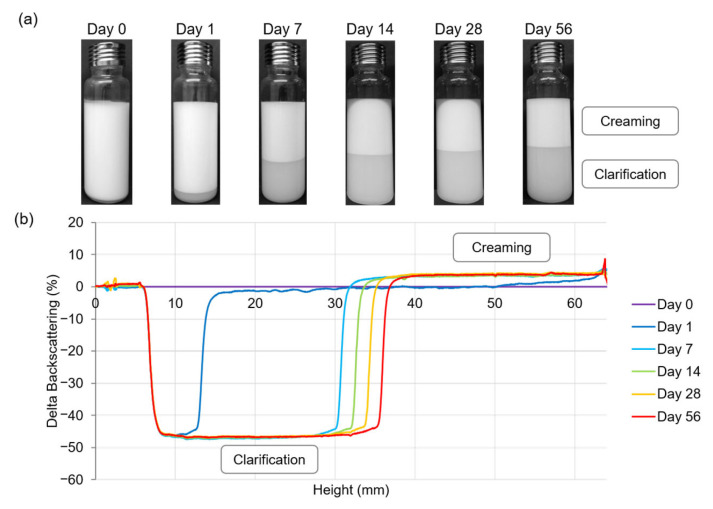
Stability evolution of a Pickering emulsion composed of 2% (*w*/*w*) CNCs and 25% (*w*/*w*) oil. (**a**) Visual appearance of emulsion from 0 to 56 days of storage at 20 °C. (**b**) Multiple light scattering analysis showing backscattering data as a function of sample height (6 to 64 mm) and of storage time (0 to 56 days).

**Figure 4 polymers-15-02783-f004:**
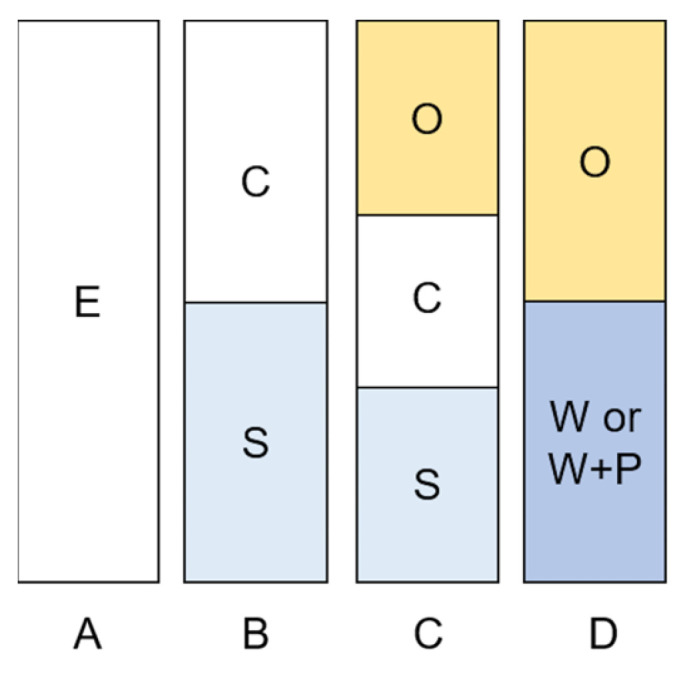
Classification based on appearance of PEs formulated with CNCs and caprylic/capric triglycerides. E: white emulsion; C: cream layer; S: serum layer; O: oil; W: aqueous phase; W + P: aqueous phase and CNCs precipitate.

**Figure 5 polymers-15-02783-f005:**
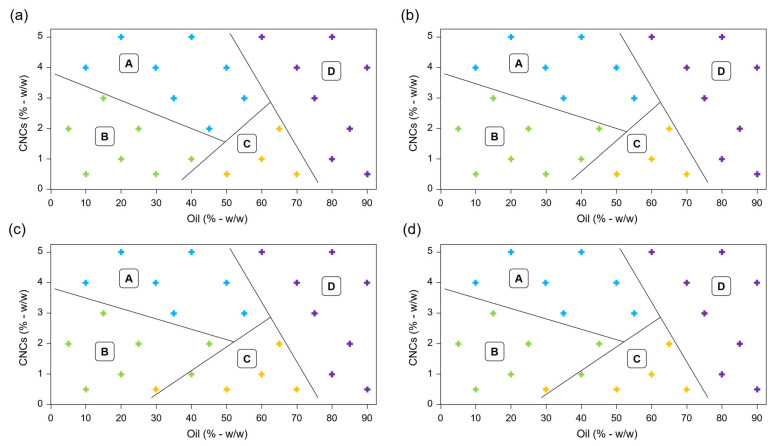
Phase diagrams of Pickering emulsions after (**a**) 7 days, (**b**) 14 days, (**c**) 28 days, and (**d**) 56 days of storage. Area defined as A, B, C, and D correspond to PEs classification as illustrated in Figure 4. Blue, green, yellow and purple crosses represent PEs belonging to categories A, B, C and D, respectively.

**Figure 6 polymers-15-02783-f006:**
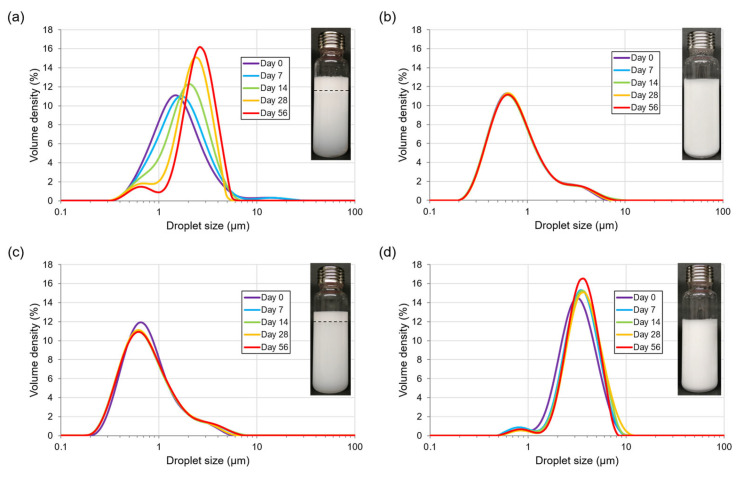
Evolution of droplet size distribution during storage (0 to 56 days) of Pickering O/W emulsions formulated with (**a**) 0.5% (*w*/*w*) CNCs and 10% (*w*/*w*) oil; (**b**) 4% (*w*/*w*) CNCs and 10% (*w*/*w*) oil; (**c**) 2% (*w*/*w*) CNCs and 5% (*w*/*w*) oil; (**d**) 4% (*w*/*w*) CNCs and 45% (*w*/*w*) oil. In inserts: photographs of emulsions after 56 days of storage with creaming indicated in dotted line for (**a**,**c**).

**Figure 7 polymers-15-02783-f007:**
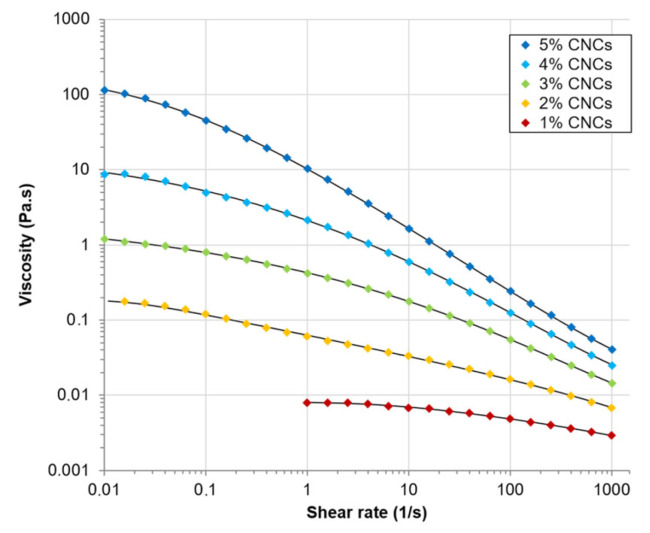
Steady-state viscosity (Pa·s) of CNCs suspensions with concentration ranging from 1 to 5% (*w*/*w*) as a function of shear rate (1/s). Black lines represent data fitted with the Carreau–Yasuda model.

**Figure 8 polymers-15-02783-f008:**
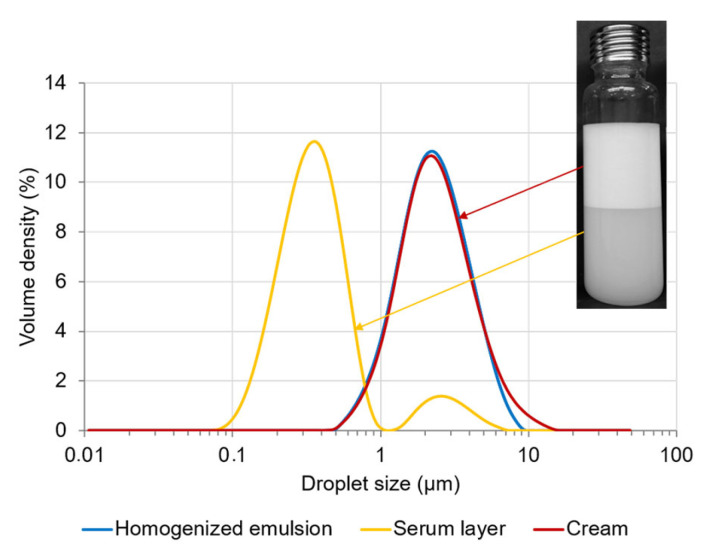
Droplet size distribution of different phases (serum layer, cream and manually homogenized total emulsion) of Pickering emulsion composed of 2% (*w*/*w*) CNCs and 25% (*w*/*w*) oil, after 56 days of storage at 20 °C.

**Figure 9 polymers-15-02783-f009:**
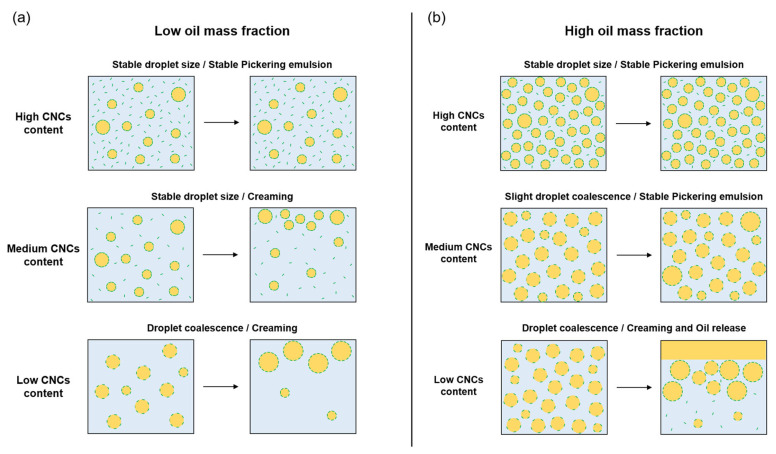
Mechanisms proposed during storage of PEs, depending on CNC content for (**a**) low oil mass fraction (approximately 5 to 25% *w*/*w*) and (**b**) high oil mass fraction (approximately 25 to 50% *w*/*w*).

## Data Availability

Not applicable.

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
