# Peer review of "Phase Diagram of Pickering Emulsions Stabilized by Cellulose Nanocrystals"

_polymers, 2023, doi:10.3390/polym15132783_

Round 1

Reviewer 1 Report

The paper presents a series of information about Pickering emulsions stabilized by cellulose nanocrystals

The paper may be of interest to the scientific community through the topic addressed.

Authors should consider the following observations:

- The ultrasonic activation process should be better presented. It is not enough to present only the frequency. It is necessary to specify other parameters such as amplitude, activation time, etc. A presentation of macroscopic images of the installation used in the research is required;

- The results obtained must be discussed in detail in order to highlight their novelty in relation to other research in the field. Thus, these results must be compared with other results obtained in other current research in order to highlight the novelty of the research carried out;

- At the end of the conclusions, future research directions should be specified

Author Response

Dear Reviewer,

First of all, we thank you for your extensive evaluation of our paper. All comments have been considered and we explain below the details of the revisions in the manuscript.

Lines 77 to 84, we completed the Introduction part, as you thought it could be improved. We added some references concerning effect of cellulose nanocrystals and oil mass fraction on Pickering emulsions formation and stability.

Lines 115 to 118, all parameters of low-frequency ultrasounds treatment are specified, as you requested. You also asked us to represent our emulsification installation. We did not add it, as we think it would not add any additional information, given that it is a commonly used method for emulsification. However, if you think it would be necessary to add a representation, we could add the figure presented in the attached file.

Lines 250 to 252 and 264 to 269, we compared our results to other research to highlight what was new in our study, as you recommended.

Lines 398 to 400 and 402 to 405, we specified future research directions in the Conclusion part, as you requested.

Best regards,

Louise Perrin, PhD Student

Reviewer 2 Report

In this work, phase diagrams of Pickering emulsions composed of cellulose nanocrystals and caprylic/capric triglycerides were investigated. The current manuscript needs minor revision before acceptance and some recommendations are listed.

 1. Why caprylic/capric triglycerides were used to stabilize PE? Does the investigation of the phase paragraph of PE composed CNCs and caprylic/capric triglycerides have a specific application or importance?

Minor editing of the English language

Author Response

Dear Reviewer,

First of all, we thank you for your extensive evaluation of our paper. All comments have been considered and we explain in this letter the details of the revisions in the manuscript.

Lines 77 to 84, we completed the Introduction part, as you thought it could be improved. We added some references concerning effect of cellulose nanocrystals and oil mass fraction on Pickering emulsions formation and stability.

Lines 97 to 99, we explained why we used caprylic/capric triglycerides in our study, as you demanded.

Lines 378 to 382, we detailed the aim of our study and explained why we designed our study with phase diagrams, as you asked.

The English language has been revised.

Best regards,

Louise Perrin, PhD Student
